# Clear Cell Renal Cell Carcinoma: From Biology to Treatment

**DOI:** 10.3390/cancers15030665

**Published:** 2023-01-21

**Authors:** Adam M. Kase, Daniel J. George, Sundhar Ramalingam

**Affiliations:** 1Mayo Clinic, Division of Hematology Oncology, Jacksonville, FL 32224, USA; 2Duke Cancer Institute, Duke University School of Medicine, Durham, NC 27710, USA

**Keywords:** clear cell renal cell carcinoma, immunotherapy, VEGF

## Abstract

**Simple Summary:**

Over the past two decades, biological discoveries have transformed the treatment strategies for renal cell carcinoma. These advances have led to the development of agents targeting pro-angiogenic pathways and the immunogenicity of renal cell carcinoma. This review will explore the biology of clear cell renal cell carcinoma and how these discoveries shaped previous therapies, current management, and future directions.

**Abstract:**

The majority of kidney cancers are detected incidentally and typically diagnosed at a localized stage, however, the development of regional or distant disease occurs in one-third of patients. Over 90% of kidney tumors are renal cell carcinomas, of which, clear cell is the most predominate histologic subtype. Von Hippel Lindau (VHL) gene alterations result in the overexpression of growth factors that are central to the pathogenesis of clear cell carcinoma. The therapeutic strategies have revolved around this tumor suppressor gene and have led to the approval of tyrosine kinase inhibitors (TKI) targeting the vascular endothelial growth factor (VEGF) axis. The treatment paradigm shifted with the introduction of immune checkpoint inhibitors (ICI) and programed death-1 (PD-1) inhibition, leading to durable response rates and improved survival. Combinations of TKI and/or ICIs have become the standard of care for advanced clear cell renal cell carcinoma (ccRCC), changing the outlook for patients, with several new and promising therapeutic targets under development.

## 1. Introduction

Kidney or renal cancer is among the most common cancers in the United States. It is estimated that over 76,000 people were diagnosed and over 13,780 died of kidney cancer in the year 2021 [1]. Approximately 75% of patients with RCC are diagnosed with clear cell (ccRCC), which typically originates in the tubular epithelium of the kidney, and can metastasize to other organs including the liver, bone, lungs, and brain [2]. Despite a lack of systematic screening, most patients are diagnosed with organ-confined RCC; only 16% of patients present with regional disease and another 16% with distant metastatic disease. Population statistics demonstrate a 70% five-year survival for those that present with regional disease and only 13% for those that present with distant metastasis [1]. Historically, cytotoxic chemotherapy had little to no activity in patients with RCC. Cytokine-based therapy with agents such as interferon-alpha and interleukin-2 were used with rare responses until the development of more targeted agents [3,4]. In 2005 and 2006, sorafenib and sunitinib were the first orally available FDA approved targeted agents targeting the VEGF receptor; over the next decade, a multitude of targeted therapies for the treatment of ccRCC entered the market (Figure 1). Since then, ICI therapy and combination therapies have become standardly used approaches to treat kidney cancer. This review will explore the biology of clear cell renal cell carcinoma and how these discoveries shaped previous therapies, current management, and future directions.

## 2. VHL/Targeted Therapies

The observation of familial syndromes of patients with similar patterns of vascular growths was perhaps the first clue into elucidating the pathogenesis of ccRCC. As early as 1894, Collins reported a case of a brother and sister with intra-ocular vascular growths, a finding later confirmed by Von Hippel and Lindau in the early 1900s [5]. It was not until the 1960s that Melmon and Rosen first reviewed the literature on this topic and developed the clinical diagnostic criteria for Von Hippel Lindau (VHL) disease, an autosomal dominant inherited disorder of neoplasms involving the central nervous system, retina, hemangioblastomas, clear cell renal cell carcinoma, and pheochromocytoma [6]. These observations were the phenotypic descriptions of what we would later learn regarding the genetic aberrations of RCC and the central role of angiogenesis in the biology of ccRCC.

In the late 1980s, Seizinger et al. mapped the loss of regions on chromosome 3p25 to be associated with sporadic and VHL forms of renal cell carcinoma [7]. The elucidation of the inactivated VHL gene on chromosome 3p25 in the year 1993 is often cited as the key finding that led to the understanding of the central biology of ccRCC. Under normal conditions, the VHL gene serves as a key tumor suppressor gene preventing cell proliferation, angiogenesis, and cell differentiation [8]. This discovery paved the way for the development of multiple therapeutic options to treat ccRCC in the following decade (Figure 1) [9]. Central to the pathogenesis of ccRCC was the understanding that an inactivated VHL gene results in upregulation of hypoxia-inducible factor (HIF), promoting the activation of pro-angiogenic pathways. Under conditions of normal oxygen tension, a normal functioning VHL protein functions in the ubiquitination and degradation of the hypoxia inducible factors (HIFs), a group of transcription factors regulating the expression of over 1000 target genes including VEGF. Under conditions of low oxygen tension, the VHL protein cannot recognize HIF-alpha as a substrate allowing the formation of heterodimers which promote the transcription of hypoxia response elements. The loss of VHL can occur through chromosomal loss, hypermethylation, or mutation, and the loss of VHL activity is estimated to occur in as high as 90% of clear cell RCC [10].

Understanding this biology has led to the development of therapeutics that target VEGF. VEGF is produced in high quantities in patients with ccRCC. One of the first drugs investigating the potential of VEGF blockade was the monoclonal antibody bevacizumab. The AVOREN trial was a multi-center randomized, double-blind, phase III trial of 649 previously untreated patients with metastatic RCC who were randomized to receive interferon alfa-2a and bevacizumab or placebo and interferon alfa-2a, a combination that resulted in a progression-free survival improvement (10.2 months vs. 5.4 months; HR 0.63, 95% CI- 0.52–0.75; *p* = 0.0001) [11].

Contemporary to the AVOREN trial, oral VEGF receptor TKIs were also under development in RCC. VEGFR is a family of receptors comprised of VEGFR1, VEGFR2, and VEGFR3, with VEGFR2 being the most biologically important receptor in terms of regulating different intracellular processes [12]. A series of oral VEGFR TKIs have now been approved for the treatment of RCC, including sorafenib (Nexavar^®^, Bayer), sunitinib (Sutent^®^, Pfizer), pazopanib (Votrient^®^, Novartis), axitinib (Inlyta^®^, Pfizer), cabozantinib (Cabometyx^®^, Exelixis), lenvatinib (Lenvima^®^, Eisai), and tivozanib (Fotivda^®^, Aveo). One of the earliest approved and previously widely used first line therapies was the multi-targeted tyrosine kinase sunitinib. In a phase III, multi-center randomized trial, 750 previously untreated patients with metastatic RCC were randomized to receive either sunitinib or interferon. The primary endpoint was progression free survival, and a statistically significant benefit was noted by 11 months with sunitinib versus 5 months with interferon (HR 0.42, 95% CI 0.32–0.54; *p* < 0.001) [13]. Pazopanib was shown to have similar efficacy in untreated metastatic RCC and became a commonly used first line standard treatment option for patients along with sunitinib for many years [14]. In a phase III, randomized trial (COMPARZ), 1110 untreated patients with clear-cell RCC were randomized to receive pazopanib or sunitinib, with the primary end point result showing that pazopanib was noninferior to sunitinib with respect to progression-free survival (HR 1.05, 95% CI 0.9–1.22). Studies also indicated the benefits of using one TKI in sequence with another TKI in patients with metastatic RCC. Although all TKIs block VEGFR1-3, they each have varying degrees of activity against these VEGFR targets, and against other pathways (Table 1) [15]. The multi-targeted nature of the TKIs has allowed for the sequential use of these agents to help overcome resistance [15].

Understanding the biological pathways of RCC and the crosstalk between biological pathways has also led to therapeutics blocking the mammalian target of rapamycin pathway (mTOR). mTOR complexes are activated in the majority of clear cell renal cell carcinomas [22]. mTOR complexes play an important role in metabolism and cell growth, as well as in modulating mitochondrial function. mTOR activation has been shown to enhance the expression of HIF-1α and HIF-2α in RCC and blocking this pathway has been shown to result in impaired expression of HIF-1α, serving as a therapeutic target [23]. The mTOR inhibitors everolimus and temsirolimus are approved in monotherapy for patients with metastatic RCC [24,25]. The combination of the VEGFR TKI lenvatinib along with everolimus has also shown to be a potent combination therapy for patients who have progressed after first line therapy for metastatic RCC [17]. In a phase II, open label, randomized, multi-center trial, 153 patients with advanced, metastatic RCC who had received a VEGF targeted therapy were treated with lenvatinib, everolimus, or lenvatinib plus everolimus. Lenvatinib plus everolimus significantly prolonged progression-free survival compared with everolimus alone (14.6 months vs. 5.5 months, HR 0.4, 95% CI 0.24–0.68; *p* = 0.0005).

Targeted therapies have left a significant impact in the treatment and prognosis of patients with metastatic RCC. Biological discoveries beginning in the mid-1990s eventually led to the approval of multiple targeted agents which improved the outlook for patients with metastatic kidney cancer. A meta-analysis was conducted on patients receiving anti VEGFR/VEGFR agents compared to those receiving placebo or interferon and it revealed a significant reduction in the risk of death 13% (HR 0.87; 95% CI, 0.8–0.95) [26]. A meta-analysis of 14,521 eligible patients including 4149 patients with metastatic renal cell carcinoma showed an improvement in overall survival in the targeted era (HR 0.87; 95% CI 0.84–0.91) including patients with clear cell histology (HR 0.76, 95% CI 0.72–0.80) [27]. The International Metastatic Renal Cell Carcinoma Database Consortium (IMDC) risk model was developed as a prognostic tool for patients treated with first-line targeted therapy for metastatic kidney cancer [28]. Based predominantly on data from patients receiving targeted therapies, this model incorporated six clinical criteria including time from diagnosis to systemic therapy< 1 year, Karnofsky performance status < 80%, corrected calcium > normal, hemoglobin < normal, neutrophil > normal, and platelet count > normal to help prognosticate survival in patients with metastatic RCC. An updated analysis and validation of this model indicated a median OS of 43.2 months in the favorable risk group (0 risk factors), 22.5 months in the intermediate risk group (1–2 risk factors), and 7.8 months in the poor risk group (3 or more risk factors) [29]. This model continues to be used widely today in clinical practice and as a predictive tool for responses to new combinations of immunotherapies. VEGFR axis has proven to be a key therapeutic target in metastatic RCC leading to improved outcomes in these risk categories. As translational work advanced, it was demonstrated that RCC has a unique immunogenicity that would forever change the treatment landscape (Figure 2).

## 3. Immunotherapy

The landscape of RCC management was forever changed with the introduction of immunotherapy into the treatment paradigm. The immunogenicity of RCC is founded in the tumor’s ability to induce an immune response that can prevent its growth [30]. The tumor microenvironment (TME) plays a pivotal role in the immunogenicity and immunoediting to escape the immune system and antitumor effects. Based on RNA and T–cell receptor sequencing, we know that the ccRCC TME has an increase in tumor infiltrating CD8^+^ T cell and macrophage populations [31]. While CD8^+^ T cell infiltration is usually associated with improved survival in other solid tumors, in ccRCC it is associated with a worse prognosis [32]. This worse prognosis is likely the result of T cell exhaustion as demonstrated by the elevated expression of immune evasive biomarkers and immunosuppressive cell infiltrates in the TME which can be explained by immunoediting [33,34]. The immunoediting hypothesis describes how the immune system protects the host from tumor formation, but also creates a tumor immunogenicity as hosts without an intact immune system were more immunogenic and “unedited” thereby promoting tumor formation [35]. Immunoediting relies on three phases, (1) Elimination via innate and adaptive immune response, (2) Equilibrium relying on adaptive immunity to keep the cancer “dormant” and (3) Escape, as a result of tumor changes or alteration in the hosts immune system [35]. These phases highlight points of therapeutic interventions in order to improve antitumor immune response by combating these suppressive and escape mechanisms.

The highly immunogenic nature of this tumor type fostered the success of immunotherapy, and its history begins with agents prior to immune checkpoint inhibitors. Therapeutics aimed at modulating the immune system were initially observed when RCC was treated with interleukin-2 (IL-2) and interferons (IFN). These signaling cytokines are instrumental in the immunologic response to neoplastic cells and involved in innate immunity (natural, immediate, nonspecific) and adaptive immunity (acquired, specific). IL-2 is key a modulator of T cells including the expansion and cytotoxicity of effector T cells and the development of memory T cells [36,37]. High-dose (HD) IL-2 was FDA approved for metastatic RCC in 1992 and was able to achieve an objective response rate (ORR) of 14% including 5% of patients achieving a complete response (CR), introducing the feasibility of a cure for a few patients with advanced disease [38,39]. Histologic investigation was performed on 388 patients treated with HD IL-2 and found patients with clear cell histology had a higher ORR when compared to other variants, 21% (30/146) vs. 6% (1/17), respectively [40]. This treatment was limited by its cytokine release syndrome toxicity encouraging alternative, and less toxic, therapeutic strategies. Type I interferons (IFN-α and IFN-β) have demonstrated direct antitumor effects and indirect antitumor effects by way of immune effector cells and vasculature [41]. IFN-α2a was FDA approved in 2009 in combination with bevacizumab for patients with untreated metastatic RCC and was shown to stimulate host mononuclear cells and enhance the expression of major-histocompatibility-complex molecules [42]. As previously mentioned, in the AVOREN trial, this combination resulted in an mPFS of 10.2 months compared with 5.4 months with bevacizumab alone (HR 0.63, 95% CI 0.52–0.75; *p* = 0.0001) [11,43]. The ORR was also improved when compared with bevacizumab alone, 31% vs. 13% (*p* < 0.001); however, a final analysis in 2010 revealed no difference in overall survival which might be the result of post-trial therapies confounding the OS analysis [43]. These agents introduced the impact that immunomodulators can have on RCC and opened the door for investigating the efficacy of immune checkpoint inhibitors.

Within the TME of ccRCC the main immune cell population is T cells in 51% of patients followed by myeloid (31%), NK cells (9%), and B cells (4%) [44]. Despite a robust T cell infiltration, RCC can proliferate and survive, in part due to T cell phenotypes causing an immunosuppressed environment or an exhausted T cell population. T cell suppression occurs as a result of regulatory CD4+ T cells which impair antitumor T cells via cytokines (IL-10, TGFβ) and immune-inhibitory receptors such as CTLA-4 [45]. Immune inhibitory receptors or “checkpoints”, located on immune cells, are necessary in maintaining tolerance to self and controlling T cell activation. These checkpoints can be utilized by tumors to evade the immune system resulting in T cell exhaustion. In addition to CTLA-4, other checkpoints include B and T lymphocyte attenuator (BTLA), T cell immunoglobulin mucin 3 (TIM3), lymphocyte-activation gene 3 (LAG3), natural killer cell receptor 2B4, and Programmed cell death-1 (PD-1) [46]. PD-1 is expressed on activated effector T cells and when bound to PD-L1 or PD-L2, it inhibits T cell activation. This negative immunoregulatory pathway has been combated with the introduction of immune checkpoint inhibitors targeting the CTLA-4 and PD-1 axis (Figure 1).

Nivolumab (Opdivo^®^, Bristol Myers Squibb), a PD-1 inhibitor, became the first immune checkpoint inhibitor approved for the management of advanced RCC in 2015. Nivolumab was compared to everolimus (Afinitor^®^, Novartis) in previously treated patients with advanced RCC and improved median overall survival from 19.6 months to 25 months (HR 0.73, 95% CI 0.57–0.93, *p* = 0.002) [47]. The ORR for nivolumab was 25% vs. 5% for everolimus and four patients (1%) receiving immunotherapy achieved a complete response (CR). In 2018, nivolumab plus ipilimumab (Yervoy^®^, Bristol Myers Squibb) became the first FDA approved immune checkpoint inhibitors in the frontline setting for patients with advanced RCC. The approval was based on CheckMate 214, when this combination was compared to sunitinib and resulted in an overall survival improvement (HR 0.63, 95% CI: 0.44–0.89, *p* = 0.0001) [48]. The 5-year update revealed a remarkable median OS of 47 months for immunotherapy vs. 26.6 months with sunitinib (HR 0.68, 95% CI 0.58–0.81) and ORR of 42% with 11% CR vs. 27% with 2% CR with sunitinib [49].

To improve the efficacy of immunotherapy, translational work investigated combining anti-VEGF agents with immunotherapy. Anti-VEGF therapy has been shown to inhibit the infiltration of suppressive immune cells including myeloid-derived suppressor cells, regulatory T cells, and macrophages suggesting that combining these agents with immune checkpoint inhibitors could result in synergistic activity [50]. Monocytic myeloid-derived suppressor cells cause immunosuppressive effects by lymphocyte nutrient depletion, the generation of oxidative stress, interfering with lymphocyte trafficking and viability, and the activation and expansion of regulatory T cells. Instead of mature myeloid cells (dendric cells, macrophages, and granulocytes), cancer cells generate activated immature cells or myeloid-derived suppressor cells (MDSCs) encouraging an immunosuppressive environment [51]. Given the immunomodulatory effects of anti-VEGF therapy, these agents were combined with immunotherapy to minimize an immunosuppressive environment. In the JAVELIN Renal 101 trial, avelumab (Bavencio^®^, Merck and Pfizer) (anti-PD-L1) was combined with the VEGFR inhibitor, axitinib, and led to FDA approval in 2019 for patients with untreated advanced RCC. When compared to sunitinib, avelumab plus axitinib resulted in a median progression free survival (PFS) improvement of 13.8 months vs. 8.4 months (HR 0.69, 95% CI 0.56–0.84, *p* < 0.001). Among patients with PD-L1 positive tumors the ORR was 55.2% (4.4% CR) vs. 25.5% (2.1% CR) for sunitinib. In KEYNOTE-426, pembrolizumab (Keytruda^®^, Merck) (anti-PD-1) was also combined with axitinib and compared to sunitinib in patients with advanced RCC. The mPFS was 15.1 months vs. 11.1 months in the sunitinib group with an ORR of 59.3% (5.8% CR) and 35.7% (1.9% CR), respectively [52]. This led to FDA approval in 2019 and after 42.8-month follow-up the mOS was 45.7 months vs. 40.1 months [53]. In 2021 FDA approved nivolumab plus cabozantinib in the frontline setting based on CheckMate9ER. Nivolumab plus cabozantinib was compared to sunitinib and showed improved outcomes with a PFS of 16.6 (immunotherapy) vs. 8.3 (sunitinib) (HR 0.51, *p* < 0.001), an ORR of 55.7% (8.0% CR) vs. 27.1% (4.6% CR) (*p* < 0.001), and a 12-month overall survival, vs. sunitinib, which was longer at 85.7% vs. 75.6% (HR of 0.60, *p* = 0.001), respectively [54]. In a phase III trial of pembrolizumab plus lenvatinib vs. lenvatinib, everolimus, or sunitinib alone, the immunotherapy combination led to an improved PFS, response rate, and overall survival leading to FDA approval in the frontline setting. The PFS was 23.9 (immunotherapy) vs. 9.2 (sunitinib), 14.7 vs. 9.2 months (HR 0.39, 95% CI 0.32–0.49, *p* < 0.001), and overall survival, vs. sunitinib, was longer with an HR of 0.66 (95% CI 0.49–0.88, *p* = 0.005) [55].

Given the success of immunotherapy in the metastatic setting, pembrolizumab was evaluated as adjuvant therapy in KEYNOTE-564 for patients who underwent nephrectomy with a high-risk for recurrence (tumor stage 2 with nuclear grade 4 or sarcomatoid differentiation, tumor stage 3 or higher, regional lymph-node disease, or stage M1 with no evidence of disease). Patients with RCC with a clear cell component were randomized to receive pembrolizumab or placebo. The disease-free survival at 2 years was longer in the immunotherapy group at 78.3% vs. 67.3% (HR 0.63, *p* < 0.0001), respectively [56,57]. The overall survival had an HR of 0.52 (95% CI 0.31–0.86, *p* = 0.0048) which has not met the planned statistical significant *p*-value and further follow-up is needed as only 33% of events needed had occurred at the time of follow-up [57]. The remarkable results of immunotherapy (Table 2) encourages further advancement in the management of RCC with investigations into new therapeutic strategies with and without immunotherapy and means of selecting patients who will achieve the greatest response.

## 4. Cytoreductive Nephrectomy

In the 1960s, there were multiple reports of metastatic RCC regression following nephrectomy prompting further investigation into its benefits [61,62,63]. This regression of metastases may be related to the primary tumors ability to alter immune cellular signaling and regulation of these pathways leading to the diversion of circulating antibodies and immune cells away from distant metastatic disease [64,65]. Primary RCC lesions are known to have a high amount of tumor infiltrating lymphocytes (TILs), however, they appear to have greater dysfunction than circulating lymphocytes providing insight into the primary tumors microenvironment’s ability to cause immune dysfunction [65]. It has been shown that RCC leads to elevated proinflammatory and T cell inhibitory cytokines such as IL-6, IL-8, IL10, granulocyte-macrophage colony-stimulating factor, and TGF-β, thereby suppressing the immune system [66]. This immune suppression was the etiology of the poor response of primary RCC tumors to immune therapy with IFN-α [67]. Therefore, removing the primary RCC tumor may lead to the removal of these immune suppressive effects allowing for the host’s immune system to invade and initiate cytotoxic effects on distant metastatic disease-causing regression.

Cytoreductive nephrectomy (CN) has also been performed to control pain, local disease, urologic symptoms, and paraneoplastic syndromes such has hypercalcemia, hepatic abnormalities (Stauffer syndrome), polycythemia, and hypertension [68,69]. CN not only led to symptom improvement, but also translated to a survival benefit in the cytokine era. In the SWOG-8949 study, patients were randomized to either interferon alpha-2b with or without cytoreductive nephrectomy. With approximately 120 patients in each arm, the median OS was higher in the nephrectomy plus interferon alpha-2b group compared with the interferon alpha-2b alone (11.1 vs. 8.1 months; *p* = 0.05). This median OS benefit was confirmed in the EORTC 30947 study when approximately 40 patients were randomized to these same treatment groups (17 vs. 7 months, HR 0.54; 95% CI, 0.31–0.94). However, in the targeted therapy era with anti-VEGFR therapy the survival benefits of CN were questioned. Two randomized trials, SURTIME and CARMENA, investigated immediate vs. delayed CN. While each of these trials have their limitations, the SURTIME trial showed the median OS was higher in the deferred CN group compared with immediate CN (32.4 vs. 15.0 months; *p* = 0.03), suggesting that CN should be offered if patients do not progress during systemic anti-VEGFR therapy [70]. The CARMENA trial showed the survival with sunitinib monotherapy was non-inferior to CN plus sunitinib (mOS 18.4 vs.13.9 months; HR 0.89; 95% CI, 0.71 to 1.10) in patients with intermediate to poor risk by the Memorial Sloan Kettering Cancer Center prognostic model [71]. In a post-hoc analysis of CARMENA in which patients were risk stratified by International Metastatic RCC Database Consortium (IMDC) criteria, patients with one IMDC criteria had a trend towards improved survival with CN plus sunitinib compared with sunitinib alone (31.4 vs. 25.2 months; HR, 1.30; *p* = 0.2). In patients with two IMDC risk factors, overall survival was significantly worse in patients who underwent CN plus sunitinib vs. sunitinib monotherapy (mOS 16.6 vs. 31.2 mo, HR 0.61; *p* = 0.015). Another subgroup analysis of CARMENA, showed that 40 patients in the sunitinib only arm who underwent a nephrectomy almost 1 year after initiating therapy had improved survival compared with those who never underwent CN (mOS 48.5 vs. 15.7 months; *p* < 0.01) [72]. With the introduction of immune checkpoint inhibitors into the treatment paradigm of RCC, the role and appropriate timing of CN is undergoing investigation. In an observation study from the international metastatic renal cell carcinoma database consortium, a multivariable analysis showed that upfront CN was associated with significantly better OS in both the immune checkpoint inhibitor patients (HR 0.61, 95% CI 0.4–0.9; *p*-0.013) and the targeted therapy patients (HR 0.72, 95% CI 0.67–0.78; *p* < 0.001 [73]. While this provides some insight, further randomized trials are needed. Ongoing prospective studies such as the NORDIC SUN trial (NCT03977571) and SWOG-1931/PROBE study (NCT04510597) will provide further evidence into the timing of CN in intermediate or poor risk patients treated with combined immunotherapy or immunotherapy plus targeted therapy combinations, respectively. With the available evidence to date, upfront CN should be considered in patients with symptomatic kidney tumors or considered in patients with favorable or intermediate disease who are candidates for oligometastatic metastasis directed therapy and are achieving no evidence of disease. Deferred CN should be considered if patients have partial or complete responses to systemic therapy. On the other hand, patients with poor risk, rapidly progressive, or high burden disease should rarely undergo CN.

## 5. Radiation Therapy

Renal cell carcinoma has historically been considered radioresistant based on in vitro studies and in clinical trials in which patients undergoing adjuvant conventional fractionated radiation did not achieve any improvement in local recurrence [74]. However, retrospective reviews of patients undergoing stereotactic ablative body radiation (SAbR) or hypofractionated radiotherapy have shown local control rates of 90–98% [74]. This control rate is likely the result of these radiotherapy techniques’ ability to deliver high doses of radiation in a single dose or a small number of fractions with high precision. While local control is feasible with radiation, an abscopal effect has been described in which localized radiation to a primary site can induce an antitumor response at distant metastatic sites which were not directly in the radiation field. Similarly to metastasis regression following CN, an abscopal effect following localized radiation to RCC has been observed in case reports [75]. In animal studies, the abscopal effect appears to be immune mediated as T cell competent mice had better disease control with lower doses of radiation compared with T cell depleted mice [76]. In other preclinical studies, radiotherapy resulted in increased tumor expression of PD-L1, tumor antigen release, improved antigen presentation, and an increase in antitumor immunity [77,78,79]. Given that preclinical studies have suggested that radiation augments the immune response, the single arm NIVES study (n = 79) investigated the effects of combining nivolumab with stereotactic body radiation therapy in advanced renal cell carcinoma who progressed after antiangiogenic therapy [77]. Unfortunately, the primary end point of improving response rates to 40% was not met as the ORR of the intent-to-treat group was 17.4%. In another trial of 25 patients, the RADVAX study, radiation (50 Gy in 5 fractions to 1–2 disease sites) was combined with dual immune check point blockade ipilimumab plus nivolumab in patients with clear cell RCC who had at least two metastatic lesions [80]. This ORR was 56% which was higher than the expected response rate of 40%, however, the median PFS was only 8.2 months which is less than the 11.6 months seen in the ipilimumab/nivolumab CheckMate-214 trial, although this is a cross trial comparison [48]. Small studies have investigated the role of radiotherapy in advanced RCC patients with oligometastatic and oligoprogressive setting (i.e., limited sites of progression). In the oligometastatic RCC setting, SAbR has been used to delay the start of systemic therapy and can control the disease for 15.2 months (95% CI, 8.8–40.1) before the initiation of systemic therapy is needed [81]. In a study of 36 patients who had oligoprogressive disease, SAbR resulted in an mPFS of 9.2 months (95% CI, 5.9–13.2) and if patients were on immunotherapy the mPFS was >28.4 months [82]. Suggesting radiotherapy could be used to maximize the delay or maximize the duration of systemic therapy. More investigation is required to solidify the timing of radiotherapy, combination strategies and patient population.

## 6. Future Directions

The management of RCC is promising with the introduction of new therapeutic agents and various combinatory therapies. Hypoxia-inducible factor-2α (HIF-2α) inhibitors are being investigated in ccRCC after belzutifan (Welireg^®^, Merck) was approved for patients with Von Hippel Lindau syndrome with RCC, CNS hemangioblastoma, and pancreatic neuroendocrine tumors based on belzutifan showing a 49% ORR of VHL patients with RCC [83]. In VHL syndrome, the tumor suppressor gene VHL is mutated causing uncontrolled cell survival and growth [84]. The loss of VHL also leads to the transcription of HIF-2α which is involved in overexpressing hypoxia-inducible genes including VEGF, the platelet-derived growth factor-β (PDGF-β), and transforming the growth factor-α (TGF-α) which are all involved in ccRCC tumorigenesis (Figure 1) [85,86]. Based on the critical role of HIF-2a in the carcinogenesis of ccRCC, multiple trials are investigating HIF-2α inhibitors as monotherapy or in combination with immunotherapy or TKIs (Table 3). These agents will likely become the next generation of therapy for patients with advanced RCC.

The response rates to immunotherapy are encouraging, however, biomarkers to predict responses are needed. Biomarker research in RCC has included techniques aimed at immunohistochemistry, tumor mutation burden (TMB), computational TME-based classification, and genomic signatures. PD-L1 expression is known to be associated with poor response to TKI therapy and associated with worse outcomes in patients with ccRCC [87,88]. Immunohistochemistry for PD-L1 on tumor infiltrating immune cells was the first biomarker assessed for immune check point inhibitors, however, studies have shown that it does not always correlate with response. When PD-L1 expression was assessed in various RCC trials including Checkmate 025 (nivolumab), Checkmate 214 (ipilimumab/nivolumab), and KEYNOTE-426 (pembrolizumab plus axitinib), the expression did not completely predict responsiveness to these checkpoint inhibitors and patients’ outcomes were predominately not dependent on PD-L1 expression [89]. TMB, defined as the total number of mutations per coding area of the tumor genome, has been investigated as a biomarker since high TMB results in the increased production of neoantigens which generates an anti-tumor immune response [90]. However, in multiple exploratory analyses, TMB has not been predictive of improved benefits in RCC patients treated with immunotherapy including ipilimumab plus nivolumab, and avelumab plus axitinib [91,92]. Zhang et al. developed a TMEscore (tumor microenvironment score), low or high, to predict responses to immunotherapy in patients with ccRCC. Using genomic data including RNA-sequencing, single-nucleotide variant data, copy number variation, miRNA, prognostic data, and infiltrating immune cells, a TMEscore was determined with a higher TMEscore (better response) being associated with memory B cells and mutation signature 5 and signature 12, which have unknown etiology annotations. While lower TMEscores were associated with genomic signature profiles 6 and 24, which are indicative of defective DNA mismatch repair, chromosomal instability, and aflatoxin exposure. As biomarker analyses become more refined, these markers will become an instrumental part of clinical practice to identify patients who will achieve the highest efficacious response.

## 7. Conclusions

Biological discoveries in RCC have translated into enormous therapeutic progress over the last two decades. These advances have led to immunotherapy becoming the cornerstone of RCC management either alone or in combination with VEGF TKIs. Clinicians now have multiple therapeutic options in the advanced RCC setting and sights are now set on the next generation of agents targeting the HIF pathway. This drug class will be a valuable option as patients who progress on VEGF TKIs, or immunotherapy, have limited therapeutic options outside these drug classes. The future is bright with more therapeutics; however, more translational and clinical studies are needed to determine when to use these agents based on biomarker analyses in order to maximize benefits and limit toxicity.

## Figures and Tables

**Figure 1 cancers-15-00665-f001:**
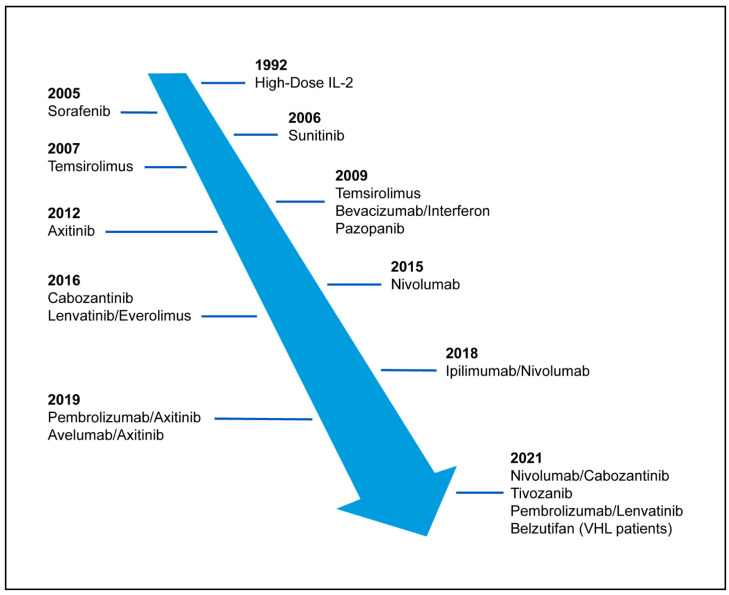
Timeline of FDA approved agents and combination treatments for metastatic renal cell carcinoma.

**Figure 2 cancers-15-00665-f002:**
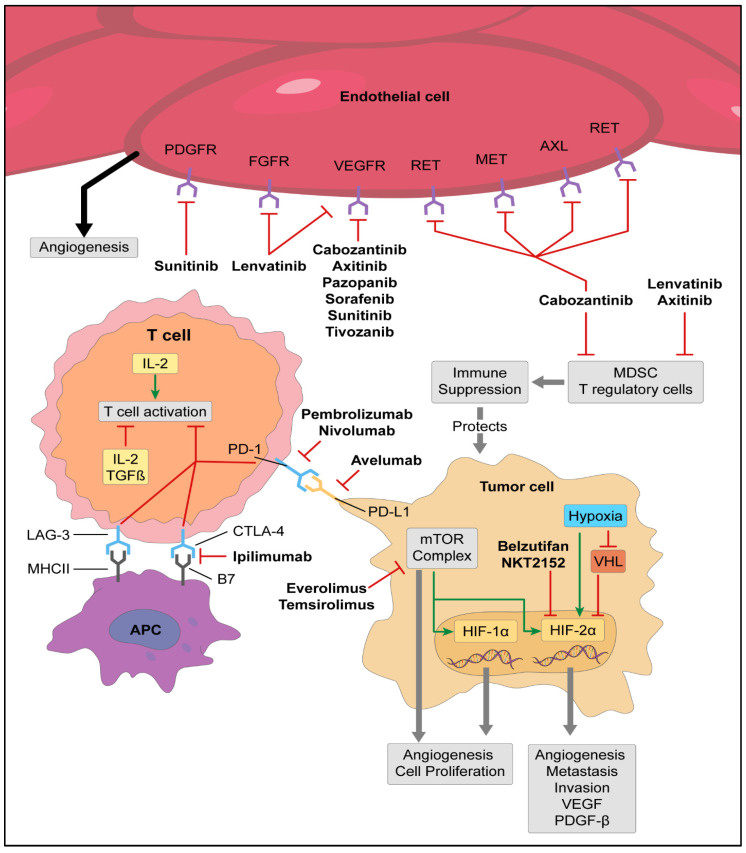
Mechanisms of action for agents used in renal cell carcinoma.

**Table 1 cancers-15-00665-t001:** FDA approved agents for advanced RCC targeting VEGF axis.

VEGFR TKI	Receptor	Trial	Efficacy
**Sorafenib** [16]	Raf-1, B-Raf, B-Raf (V599E), VEGFR, PDGFR, c-Kit, RET	Phase III RCT, sorafenib vs. placebo, previously untreated	PFS 5.5 months vs. 2.8 months (HR 0.44, 95% CI 0.35–0.55, *p* < 0.01)
**Sunitinib** [13]	c-Kit, FLT3, PDGFRβ	Phase III RCT, sunitinib vs. interferon, previously untreated	PFS 11 months vs. 5 months (HR 0.42, 95% CI 0.32–0.54; *p* < 0.001)
**Lenvatinib** [17] **(combined with everolimus)**	PDGFRα, PDGFRβ, FGFR1	Phase II RCT, lenvatinib + everolimus vs. everolimus, previously treated	PFS 14.6 months vs. 5.5 months (HR 0.4, 95% CI 0.24–0.68; *p* = 0.0005)
**Cabozantinib** [18]	c-MET, AXL, RET, KIT, FLT3, TRKB, Tie-2	Phase III RCT, cabozantinib vs. everolimus, previously treated	PFS 7.4 months vs. 3.8 months (HR 0.58; 95% CI 0.45–0.75, *p* < 0.001)
**Axitinib** [19]	PDGFRα, PDGFRβ, Kit, BCR-ABL1	Phase III RCT, axitinib vs. sorafenib, previously treated	PFS 6.7 months vs. 4.7 months (HR 0.665, 95% CI 0.544–0.812, *p* < 0.0001)
**Pazopanib** [20]	PDGFR, FGFR, c-Kit	Phase III RCT, pazopanib vs. placebo, previously untreated and cytokine pre-treated	PFS 9.2 months vs. 4.2 months (HR 0.46, 95% CI 0.34–0.62, *p* < 0.001)
**Tivozanib** [21]	Predominantly VEGFR 1–3	Phase III, RCT, tivozanib vs. sorafenib, previously treated	PFS 5.6 months vs. 3.9 months (HR 0.73, 95% CI 0.56–0.93, *p* = 0.016)

**Table 2 cancers-15-00665-t002:** FDA approved immunotherapy in patients with RCC. n/a, not available.

Immunotherapy	Immunotherapy Target	Study	ORR	PFS	OS (Months)
**High Dose IL-2** [4,39]	T cells	Phase 2 trial in advanced RCC and retrospective review	14–20% (8–9% CR)	n/a	19
**IFN-α2a** [58]	Immune cells	Prospective trial in advanced RCC	10% (1% CR)	n/a	11.4
**IFN-α2a plus Bevacizumab** [11,43]	Immune cells	Phase 3 trial in untreated mRCC combination vs. IFN-α2a monotherapy	31% vs. 13%	10.5 vs. 5.4 (*p* = 0.0001)	23.3 vs. 21.3 (ns)
**Nivolumab** [47]	PD-1	Phase 3 previously treated mRCC vs. everolimus	25% vs. 5% (1% CR Nivolumab)	4.6 vs. 4.4 (ns)	25 vs. 19.6 (HR 0.73, 95% CI 0.57–0.93, *p* = 0.002)
**Ipilimumab plus Nivolumab** [48,49]	CTLA-4/PD-1	Phase 3 untreated advanced ccRCC vs. sunitinib	42% vs. 27% (11% CR Ipilimumab/ Nivolumab)	11.6 vs. 8.4 (ns)	47 vs. 26.6 HR 0.68 (95% CI 0.58–0.81)
**Avelumab plus Axitinib** [59,60]	PD-L1	Phase 3 untreated advanced RCC vs. sunitinib	55.2% vs. 25.5% (4.4% CR Avelumab/Axitinib)	13.8 vs. 8.4 (*p* < 0.001)	19.3 vs. 19.2 (ns)
**Pembrolizumab plus Axitinib** [52,53]	PD-1	Phase 3 untreated advanced RCC vs. sunitinib	59.3% vs. 35.7% (5.8% CR Pembrolizumab/ Axitinib)	15.1 vs. 11.1	45.7 vs. 40.1 (HR, 0.73; 95% CI 0.60–0.88, *p* < 0.001)
**Nivolumab plus Cabozantinib** [54]	PD-1	Phase 3 untreated advanced RCC vs. sunitinib	55.7% vs. 27.1% (8.0% CR Nivolumab/ Cabozantinib)	16.6 vs. 8.3	12-month OS 85.7% vs. 75.6% (HR of 0.60, *p* = 0.001)
**Pembrolizumab plus Lenvatinib** [55]	PD-1	Phase 3 untreated advanced RCC vs. sunitinib	71% vs. 36% (16% CR Pembrolizumab/ Lenvatinib)	23.9 vs. 9.2	HR of 0.66 (95% CI 0.49–0.88, *p* = 0.005)
**Pembrolizumab** [56,57]	PD-1	Phase 3 adjuvant therapy after nephrectomy in high-risk RCC vs. placebo	n/a	n/a	2-year DFS 78.3% vs. 67.3% (HR 0.63, *p* < 0.0001); OS not mature

**Table 3 cancers-15-00665-t003:** Ongoing Clinical Trials of Hypoxia-Inducible Factor-2α.

Intervention	Target	Trial Type	Estimated Completion Date	Identifier	Status
**Belzutifan (standard dose vs. high dose)**	HIF-2α	Phase 2	October 2025	NCT04489771	Active, not recruiting
**Belzutifan vs. everolimus**	HIF-2α	Phase 3	September 2025	NCT04195750	Active, not recruiting
**Belzutifan plus lenvatinib vs. cabozantinib**	HIF-2α	Phase 3	December 2024	NCT04586231	Recruiting
**Belzutifan + pembrolizumab vs. pembrolizumab + placebo**	HIF-2α	Phase 3	January 2030	NCT05239728	Recruiting
**Pembrolizumab + belzutifan + lenvatinib** **or** **pembrolizumab/quavonlimab + lenvatinib ** **vs. pembrolizumab + lenvatinib**	HIF-2α	Phase 3	October 2026	NCT04736706	Recruiting
**NKT2152**	HIF-2α	Phase 1/2	September 2026	NCT05119335	Recruiting
**Belzutifan + cabozantinib**	HIF-2α	Phase 2	August 2025	NCT03634540	Recruiting

## Data Availability

Not applicable.

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
