# Peer review of "Clear Cell Renal Cell Carcinoma: From Biology to Treatment"

_cancers, 2023, doi:10.3390/cancers15030665_

Round 1

Reviewer 1 Report

Authors have done an excellent job in providing a comprehensive overview of clear cell RCC biology with development of treatment and outcomes.

Authors have attempted to describe in detail the history and sequence of events which lead to development of diagnostic criteria of VHL disease and later how it played a central role in biology of ccRCC. They further describe how understanding the biology of disease lead to developments of targeted therapies including VEGF TKIs, immunotherapy and HIF2 alpha inhibitors. So topic is relevant and interesting.

Review articles discussing treatment landscape of ccRCC have been published. Authors here are not only discussing the treatment landscape by providing information on various trials but also emphasize on the pathways and mechanism of action.

Paper is very well written and easy to read. Authors provide succinct conclusions summarizing all the data. They discuss the future directions, updates on biomarkers and its limitations.

Author Response

Reviewer 1

No suggestions were made by review 1. Thank you for your review.

Reviewer 2 Report

In this article, authors reviewed the biological principles of the management of clear cell RCC. They also  discussed the contemporary systemic management of clear cell RCC. This is a well-written, well researched and well-referenced articles. Figures and tables nicely complemented the text.

Recently, there have been a number of very well done review articles published on RCC focusing on similar topics (Tran & Ornsetin, J of Oncology Practice, 2021, McKay et al, J of clinical oncology, 2018, Kotecha, et al, Nature Oncol 2019, etc.). I think readers will find it more useful and interesting, if authors expand on some evolving and unresolvednissues. I have few comments and suggestions.

1.     Authors did one paragraph on biomarkers in ‘future direction’ section. Biomarkers in RCC deserves a separate and expanded section, synthesizing data available in the contemporary translational and clinical literature.

2.     Data are  currently inconclusive on the clinical utility of cytoreductive nephrectomy in mRCC patients.  Authors may consider including a section on biological principle of cytoreductive nephrectomy in mRCC patients, specifically in the era of targeted and check-point inhibitor immunotherapy .

3.     Traditionally, RCCs are not considered radiosensitive, and radiation is primarily used for palliation. There are evolving data of usefulness of radiation in patients with  oligo-metastatic RCCs. This approach may delay the initiation of systemic therapy. Readers might find it interesting to  know the biologic rationale of this approach.

4.     I would suggest authors use the term check-point inhibitor  immunotherapy (CPI) in-stead of immunotherapy (section 3). Authors discussed the  biological basis very well. But many of the clinical questions remain unresolved that include sequencing of available systemic therapies (particularly later line TKIs), optimal duration and re-challenge CPI at progression. There are also no guidance regarding the role of continued VEGF therapy after stopping CPI in patients, who responded to initial combination therapies.  Authors may discuss some of these issues while discussing contemporary systemic therapies.

5.     Minor comment: authors description of the objective of the article in ‘Simple Summary Section’ and ‘Introduction’ should be consistent.

Author Response

Reviewer 2

  • Authors did one paragraph on biomarkers in ‘future direction’ section. Biomarkers in RCC deserves a separate and expanded section, synthesizing data available in the contemporary translational and clinical literature.
  • Thank you for your suggestion. We agree biomarkers deserve to be highlighted, however, given that most of the data has not been confirmed and is still under investigation, we feel the biomarkers would be best served under future directions.

  • Data are currently inconclusive on the clinical utility of cytoreductive nephrectomy in mRCC patients.  Authors may consider including a section on biological principle of cytoreductive nephrectomy in mRCC patients, specifically in the era of targeted and check-point inhibitor immunotherapy.
    • We have included a separate section on cytoreductive nephrectomy. Thank you for your suggestion as we feel this has significantly strengthened the paper and provided a more comprehensive review. (Section 4, line 273)

  • Traditionally, RCCs are not considered radiosensitive, and radiation is primarily used for palliation. There are evolving data of usefulness of radiation in patients with  oligo-metastatic RCCs. This approach may delay the initiation of systemic therapy. Readers might find it interesting to  know the biologic rationale of this approach.
    • We have included a separate section on radiotherapy. Thank you for your suggestion as we feel this has significantly strengthened the paper and provided a more comprehensive review. (Section 5, line 333)

  • I would suggest authors use the term check-point inhibitor  immunotherapy (CPI) in-stead of immunotherapy (section 3). Authors discussed the biological basis very well. But many of the clinical questions remain unresolved that include sequencing of available systemic therapies (particularly later line TKIs), optimal duration and re-challenge CPI at progression. There are also no guidance regarding the role of continued VEGF therapy after stopping CPI in patients, who responded to initial combination therapies.  Authors may discuss some of these issues while discussing contemporary systemic therapies.
    • Thank you for your suggestion we have exchanged the term “immunotherapy” for check-point inhibitor immunotherapy where we felt this was appropriate to provide more clarity. (Section 3, Line 152)
  • Minor comment: authors description of the objective of the article in ‘Simple Summary Section’ and ‘Introduction’ should be consistent.
    • This has been updated. Thank you for the suggestion. (Line 44)